# Pool Boiling Amelioration by Aqueous Dispersion of Silica Nanoparticles

**DOI:** 10.3390/nano11082138

**Published:** 2021-08-22

**Authors:** Sayantan Mukherjee, Naser Ali, Nawaf F. Aljuwayhel, Purna C. Mishra, Swarnendu Sen, Paritosh Chaudhuri

**Affiliations:** 1Thermal Research Laboratory (TRL), School of Mechanical Engineering, Kalinga Institute of Industrial Technology, Bhubaneswar 751024, India; 1881148@kiit.ac.in (S.M.); pcmishrafme@kiit.ac.in (P.C.M.); 2Nanotechnology and Advanced Materials Program, Energy and Building Research Center, Kuwait Institute for Scientific Research, Safat 13109, Kuwait; nmali@kisr.edu.kw; 3Mechanical Engineering Department, College of Engineering and Petroleum, Kuwait University, P.O. Box 5969, Safat 13060, Kuwait; 4Department of Mechanical Engineering, Jadavpur University, Kolkata 700032, India; swarnendu.sen@jadavpuruniversity.in; 5Institute for Plasma Research (IPR), Bhat, Gandhinagar 382428, India; paritosh@ipr.res.in; 6Homi Bhabha National Institute, Anushaktinagar, Mumbai 400094, India

**Keywords:** nanofluids, silica, heat transfer, PBHTC, CHF

## Abstract

Non-metallic oxide nanofluids have recently attracted interest in pool boiling heat transfer (PBHT) studies. Research work on carbon and silica-based nanofluids is now being reported frequently by scholars. The majority of these research studies showed improvement in PBHT performance. The present study reports an investigation on the PBHT characteristics and performance of water-based silica nanofluids in the nucleate boiling region. Sonication-aided stable silica nanofluids with 0.0001, 0.001, 0.01, and 0.1 particle concentrations were prepared. The stability of nanofluids was detected and confirmed via visible light absorbance and zeta potential analyses. The PBHT performance of nanofluids was examined in a customized boiling pool with a flat heating surface. The boiling characteristics, pool boiling heat transfer coefficient (PBHTC), and critical heat flux (CHF) were analyzed. The effects of surface wettability, contact angle, and surface roughness on heat transfer performance were investigated. Bubble diameter and bubble departure frequency were estimated using experimental results. PBHTC and CHF of water have shown an increase due to the nanoparticle inclusion, where they have reached a maximum improvement of ≈1.33 times over that of the base fluid. The surface wettability of nanofluids was also enhanced due to a decrease in boiling surface contact angle from 74.1° to 48.5°. The roughness of the boiling surface was reduced up to 1.5 times compared to the base fluid, which was due to the nanoparticle deposition on the boiling surface. Such deposition reduces the active nucleation sites and increases the thermal resistance between the boiling surface and bulk fluid layer. The presence of the dispersed nanoparticles caused a lower bubble departure frequency by 2.17% and an increase in bubble diameter by 4.48%, which vigorously affects the pool boiling performance.

## 1. Introduction

Pool boiling heat transfer has emerged as a popular choice of research interest due to its variety of applications in heat transfer and energy sectors. In the pool boiling heat transfer process, the majority of industrial appliances are operated under the nucleate boiling region because of the high heat transfer coefficient (HTC) at low wall super heat condition. Moreover, the CHF is also regulated continuously in order to avoid any physical damage to boiling surface such as burn out or meltdown. Hence, augmentation in pool boiling heat transfer coefficient (PBHTC) as well as the critical heat flux (CHF) enhancement will be of a great commitment to achieve a higher heat transfer rate, improve energy conversion efficiency, and avoid boiling crisis.

Many boiling enhancement research studies to date have focused on boosting nucleate boiling heat transfer by increasing the effective surface area, nucleation sites, bubble departure frequency, and decreasing ONB (onset of nucleate boiling). However, traditional liquids, due to their inherent poor thermal properties, often fail to encounter very high heat flux, and therefore causing equipment damage and several accidents [1]. It was reported in the literature that the heat transfer potential of the conventional fluids can be increased several times by dispersing metallic and non-metallic nano-sized particles within these liquids [2,3]. This new class of working fluid is popularly known as nanofluid [4]. The use of nanofluids have shown tremendous enhancements in PBHTC and CHF [1,5,6].

Researchers have examined the pool boiling improvement and CHF enhancement by dispersing different nanoparticles in a variety of working fluids. Very recently, Mehralizadeh et al. [7] investigated the pool boiling performance of dispersed SiO_2_, TiO_2_, and their hybrids in water, ethylene glycol (EG), and water–EG mixtures. The hybrid nanofluids have shown a better performance compared to dispersions that were made of pure nanoparticles. The researchers have noticed that the nanoparticle concentration had a strong influence on the pool boiling heat transfer performance. Furthermore, the hybrid nanofluids at 50:50 particle ratios showed the highest PBHTC enhancement. Among the three concentrations of 0.01%, 0.03%, and 0.05%, the highest concentration was demonstrated to be optimum for the pool boiling heat transfer performance. They also reported that switching from a flat surface to a one with circular channels and intersecting lines (CC-IL) could further improve the HTC significantly with all nanofluids.

Kamel et al. [8] demonstrated the pool boiling performance of water-based dilute cerium oxide nanofluids with different volume concentrations that ranged from 0.001 to 0.04%, and they noticed significant enhancements in PBHTC and CHF. For volume concentrations of 0.007% and 0.004% (at low heat flux), the highest PBHTC ratio for cerium oxide-based water nanofluids was found to be around 1.7 and 1.6, respectively. Moreover, the surface roughness, wettability, and capillary wicking heavily influenced the performance of the pool boiling process. The visualization study revealed that bubble size increased with the applied heat flux in the base fluid as well as in the nanofluids. However, at higher heat flux, the bubble growth is more present in the base fluid compared to the nanofluids, therefore resulting in uneven bubble tower formation.

Safaei et al. [9] studied the nucleate pool boiling heat transfer of aqueous glycol alumina nanofluids as a coolant around a horizontal rod heater. They applied heat flux in the range of 0–90 kW/m^2^, kept the concentration of nanofluid at 1%, and used a volumetric concentration of 0–40% of the heavier component. They identified a natural convection zone and a nucleate boiling zone with bubble production and bubble interactions for heat transfer. The authors reported that the PBHTC of the binary mixture decreases with the increase in ethylene glycol in the proportion. Such decrease is attributed to the wide difference in the boiling points of the constituent fluids of the binary mixture, which developed a concentration gradient between the vapor–liquid interfaces. Furthermore, the water diffuses into the gas–liquid interface and evaporates on the interface as a result of such a gradient, while glycol molecules diffuse in the opposite direction. As an outcome, the developed bubbles around the heater mostly originated from water. The aforementioned has shown to develop a thick layer of heavy component on the heating portion, which therefore increases the heat resistance as well as the mass transfer resistance. Hence, a substantial reduction in bubble formation near the surface occurs, and the PBHTC value consequently reduces.

Li et al. [10] studied the transient effect on pool boiling and particulate deposition of CuO–water nanofluids with 1000 min of continuous heater operation. The heat transfer coefficient showed an enhancement at the start of the tests due to the higher thermal conductivity of the nano-suspension and the occurrence of various micro-scale phenomena such as Brownian motion and the thermophoresis effect. Throughout the 1000 min of testing, the heat transfer coefficient was found to reduce as a result of fouling formation on the boiling surface. The thermal resistance of the fouled layer, heat accumulation, decrease in surface roughness value, and suppression of the rate of bubble formation were found to be the primary contributors to the system’s transient thermal performance. They also discovered that the thermal resistance due to fouling lowers with the reduction of the surface roughness.

Mukherjee et al. [11] investigated the pool boiling performance of water-based Al_2_O_3_ and TiO_2_ nanofluids with 0.01–1% weight concentrations. They observed enhancements in PBHTC of 36.6% and 34% with Al_2_O_3_ and TiO_2_ nanofluids, respectively. As for the CHF, the enhancement was found to be 56.15% and 53.12% using Al_2_O_3_ and TiO_2_ suspensions, respectively. For the nanofluids above 0.1%, they showed significant decrease in PBHTC and CHF due to the particle deposition. Furthermore, the authors also reported an increase in the wettability and surface roughness, which correspondingly enhanced the pool boiling heat transfer performance.

Rahimian and Kazeminejad [12] studied the boiling and quenching of a steel rod 80 mm long and 15 mm diameter in SiO_2_ and TiO_2_ nanofluids at 0.01% weight concentration. They found that quenching and boiling performance increased significantly as a result of using nanofluids. The scholars reported a maximum CHF enhancement of 120% with both SiO_2_ and TiO_2_ nanofluids. Moreover, the cooling duration of the steel rod showed a reduction of 50% as an outcome of the nanoparticle deposition.

Sezer et al. [13] explored the pool boiling heat transfer performance in colloidal dispersion of nano-sized carbon black particles at three weight concentrations, namely 0.001%, 0.005%, and 0.01%. They reported 190.5% enhancement in PBHTC and 67.8% enactment in CHF with 0.01% weight concentration. They also described a nine-fold increase in the roughness of the boiling surface due to nanoparticle disposition, which provided a better heat transfer area and active nucleation site density, thus contributing toward enhancing the pool boiling mechanism.

Li et al. [14] introduced CuO–water nanofluids on a finned boiling surface for 1000 min. Their results indicated that the developed fins reduced the rate of fouling of the nanoparticles on the boiling surface, such that the surface modified with the fins with more towering height and narrow width achieved the greatest thermal performance in terms of PBHTC enhancement. Furthermore, the fouling thermal resistance was discovered to exhibit asymptotic behavior while creating three distinct areas of genesis, growth, and equilibrium. This was attributed to an increase in the system’s specific surface area and thermal performance. Finally, they concluded that the inclusion of the fins was promising in terms of enhancing the pool boiling performance in comparison to the plain surface.

Akbari et al. [15] investigated the pool boiling heat transfer of dispersed graphene as well as functionalized graphene in water nanofluids over a flat plate. The PBHTC and CHF of nanofluids were enhanced with particle loading. Functionalized graphene nanofluids showed better boiling performance than the pure graphene-based nanofluids. It was observed that more than 72% enhancement in CHF can be achieved with functionalized graphene nanofluids containing as low as 0.01% mass concentration. Moreover, they reported that the enhanced thermal conductivity of nanofluids was one of the main reasons behind the pool boiling improvement. Searching for the optimum nanoparticles size, Norouzipour et al. [16] conducted an experimental study on the PBHTC measurement of silicon oxide nanofluids with spherical nanoparticles that had three different average apparent particle sizes of 11, 50, and 70 nm. The boiling tests were performed over a flat copper surface with a variety of dispersed nanomaterial concentration, i.e., from 0.01–1.0 vol%. Their findings demonstrate that PBHTC increased with the increase in the loaded particles size. The highest value of PBHTC was obtained with the particle of 70 nm in size and 0.1% volume concentration.

Etedali et al. [17] studied the effect of different surfactants on the pool boiling performance of silica-deionized (DI) water nanofluids. Silica suspensions were prepared by adding non-ionic polysorbate 20 (Ps20), cationic cetyltrimethylammonium bromide (CTAB), and anionic sodium lauryl sulfate (SLS) surfactants. The pool boiling heat transfer was performed with 0.01–1.0% volume concentration over a copper surface. The addition of surfactants was shown to assist the heat transfer mechanism along with reducing the surface tension. As a result, the PBHTC improved because of the easier separation of bubbles from the boiling surface. The ranking of surfactants based on their impact on PBHTC was SLS > CTAB > Ps20. In other words, the lowest to highest improvements in PBHTC were found for non-ionic followed by the cationic and anionic yields. Recently, a similar type of work was presented by Tian et al. [18]. The authors used sodium dodecyl sulfate (SDS), CTAB, and Ps20 surfactants with silica–DI water nanofluids for pool boiling. They have reported that adding surfactants to the base fluid alone can enhance the pool boiling performance. From their study, the boiling heat transfer coefficient (BHTC) of nanofluids with different surfactants can be ranked in a decreasing order as anionic surfactant (SDS), cationic (CTAB), and non-ionic (PS20). This finding is similar to the result that was presented in the previous study (i.e., reference [17]). The impact of surface roughness after boiling heat transfer of nanofluids with those surfactants are in the increasing order of SDS > Pd20 > CTAB.

Lobasov et al. [19] reported the effect of surfactant on the CHF of SiO_2_ in water nanofluids with 0.1% volume concentration. They used xanthan gum and polyacrylamide as surfactants, which concentration was from 10 to 200 mg/L. Both xanthan gum and polyacrylamide improved the CHF even without the dispersed nanomaterials. The CHF improvements with xanthan gum and polyacrylamide were higher than pure water by 32% and 61%, respectively. On the other hand, the CHF values for nanofluids containing xanthan gum and polyacrylamide showed an increase of 13% and 36%, respectively, compared to those fabricated without the surfactants. Furthermore, the surfactant-based nanofluids have also shown an improvement in CHF by 200% and 263% over that of pure water when xanthan gum and polyacrylamide where employed, respectively.

Sarafraz et al. [20] presented the pool boiling result of nanofluids containing ZnO dispersed in an EG–water mixture (40:60 volume ratio) near the critical heat flux region. The PBHTC was shown enhance with the increase in mass concentration (i.e., 0.01–0.4%) and heat flux (i.e., 0–1530 kW/m^2^). However, further increasing the mass concentration was shown to reduce PBHTC significantly due to the particle deposition on the hot surface. They also concluded that the deposited layer has promoted the capillary wick and thus would attract more liquid over the examined hot surface.

Fan and Li [21] demonstrated that the length and thickness of carbon nanotubes (CNTs) influence the pool boiling performance in the quenching operation. A higher quenching rate was observed due to pool boiling and CHF enhancement in water by the dispersion of CNTs. A 60% improvement in CHF was obtained by dispersing longer and thicker CNTs. They stated that enhancement in pool boiling was attributed to the development of a micro-porous layer of deposited CNTs on the quenched surface. Therefore, it renders the better heat transfer surface and roughness enhancement of the quenching surface. As a result, heat transfer improvement was achievable.

Dogan Ciloglu [22] boiled silica nanofluids over a hemispherical heating surface under atmospheric pressure and saturated conditions. The PBHTC was shown to decrease with the increase in dispersed particles concentration. In addition, the deposition of nanoparticles created an extra thermal resistant layer and hence modified the boiling surface. In return, it reduced the boiling performance and PBHTC deterioration. As for the CHF, it was enhanced for all concentrations as compared to water, and a 45% improvement over the base fluid was obtained with only 0.1% volume concentration.

The effect of wettability on the boiling heat transfer of nanofluids was for the first time presented by Quan and Cheng [23]. They demonstrated the mechanism of wetting by visualization of the bubble dynamics in nanofluids containing hydrophilic nanoparticles and nanoparticle deposition on the heater surface. The scientists concluded that the moderately hydrophilic nanoparticles promote pool boiling heat transfer in a nucleate boiling regime by two processes. The first was by the reduction in bubbles’ interfaces, which causes the bubbles’ coalescence and the generation of smaller bubbles in size. As for the second, the enhancement of the heater surface roughness can provide more active nucleate cavities for a higher heat transfer rate, therefore leading to an enhancement in the PBHTC and CHF. Moreover, strongly hydrophilic particles inhibit heat transfer by developing a thick layer nanoparticle on the heater surface, and PBHTC deteriorated.

Recently, Gimeno-Furio et al. [24] reported the effect of aqueous carbon nanohorn pool boiling on the wettability and optical properties of different metal surfaces. They found that the deposited thin layer of nanoparticles during boiling could alter the wettability and optical properties of these metallic surfaces.

Salimpour et al. [25] examined the pool boiling characteristics of water-based ferrous oxide nanofluids with 0.05 and 0.4 volume concentrations on a flat copper wall. They developed an empirical correlation to determine the boiling coefficient (*C_sf_*) values using a neural network based nonlinear regression approach. The *C_sf_* was quantified in terms of surface roughness, concentration, and boiling heat transfer. They also added that the boiling performance of nanofluids decreased with particle loading due to the deposition of nanoparticles on the heater surface.

Goodarzi et al. [26] studied the boiling of graphene nanopallets nanofluids flow in 0.025, 0.05, and 0.1% weight fractions over a hot copper disk. They reported an increase in BHTC with the increase in both heat flux and flow rate. A maximum BHTC value of 17.4 kW/m^2^K was achieved when a 10,950 Reynolds number was employed. The inclusion of GNPs increased the surface’s fouling heat resistance. In another study, Goodarzi et al. [27] investigated the boiling heat transfer performance of graphene oxide nanoplatelets (GONP) nano-suspensions of water–perfluorohexane (C6F14) and water-n-pentane flowing inside an annular heat exchanger. The results indicated that raising the heat flux, GONP weight concentration, and working fluid temperature increased the BHTC of the system. Furthermore, the presence of GONPs was shown to enhance both friction forces and viscosity. The BHTC of the system was more significant with the nanofluids than the base fluid in all the experimental runs.

According to the above-mentioned literature, nanoparticles have a notable impact on the PBHTC and CHF enhancements. However, these studies are more orientated toward metallic and metal oxide nanoparticles with very few focusing on the other types of nanomaterials, such as silica nanoparticles. Silicon oxide or silica is a cost-effective material, which can provide a good amount of effective thermal property enhancement in base fluids [28,29]. Due to its low molecular weight, silica is considered to be lighter than most prevalent nanomaterials. As such, it can stay in a dispersed phase for longer time if well-handled because gravitational force will have less effect on it. Furthermore, it is evident from the previously mentioned literature that most of the pool boiling studies were performed on flat copper surfaces. However, investigating boiling on flat stainless steel (SS) surfaces did not receive a similar amount of attention, although a significant enhancement on pool boiling was previously reported with the flat steel plate [11]. In addition, the effects of wettability and capillary wicking using nanofluids have not been addressed fairly as well. Considering all these characteristics and scientific gaps, it has driven the conduction of the in-hand research work. As such, this study aims to investigate the pool boiling and CHF of water based SiO_2_ nanofluids over a flat rectangular stainless steel heating surface. This was done in order to achieve the following objectives listed below:To find the boiling heat transfer characteristics of silica nanofluids.To find the effect of silica nanoparticles on PBHTC and CHF in nanofluids.To find the effect of wettability, surface roughness, and capillary wicking on pool boiling heat transfer.To estimate the bubble frequency and bubble diameter during pool boiling of nanofluids.To interpret the pool boiling mechanisms using the experimental outcomes.

## 2. Materials and Methods

### 2.1. Starting Material

Spherical silica nanoparticles of 99.5% purity were purchased from Sisco Research Laboratories (SRL) Ltd., Bangalore, India, for use as the feedstock nanomaterial for producing the nanofluids. The average particle size was reported as 15 nm by the manufacturing company. Deionized (DI) water was supplied at 20 °C by an Elga PR030BPM1-US Purelab Prima 30 water purification system (High Wycombe, UK). The DI water was later used as the nanofluids base fluid after adjusting its pH value 7. This was done by adding sodium hydroxide solution of type 1.09956. Titrisol^®^ that was supplied by SIGMA-ALDRICH Inc. (St. Louis, MO, USA), while stirring and monitoring the pH changes of the liquid using a PHC20101 Intellical gel filled Ph electrode connected to a calibrated HACH company HQ11D portable pH meter device (Loveland, CO, USA). Moreover, the calibration of the pH device was performed using three commercial calibration fluids (supplied by Metrohm USA Inc., Tampa, FL, USA) of pH 4, 7, and 10, respectively. It is important to note that the accuracy of the pH device used was ±0.002 pH, as provided by the manufacturer.

### 2.2. Nanopowder Characterization

The as-received nanopowder was characterized using a JEOL JSM-IT700HR field emission scanning electron microscopy (FE-SEM) (JEOL Ltd., Tokyo, Japan) and its energy-dispersive X-ray spectroscopy (EDS) system (JEOL Ltd., Tokyo, Japan) along with the X-ray diffraction (XRD) device and particle size analyzer. These tests were used to determine the morphology of the particles (i.e., shape), particles’ average size, nanoparticle structure and composition, and impurities within the as-obtained nanomaterial. For the SEM images, two micrographs were recorded at different magnifications (i.e., 50× and 100× magnifications). The aforementioned was performed using the SEM device secondary electron mode at a distance of 5.6 mm from the targeted sample and with an accelerating voltage of 10 kV to reduce possible damages to the powder. Furthermore, the EDS analysis was performed by switching to its mode using the device operating software, InTouchScope 1.12. By using the previous system, elemental wt % and mapping were obtained. Furthermore, XRD analysis was performed using a 9 kW Rigaku SmartLab system (Rigaku Corporation, Tokyo, Japan) along with its SmartLab Guidance software. A CuKα X-ray source was used with a diffraction angle of 2θ and an incidence beam step of 0.02° to determine the Bragg’s peaks from the examined as-received sample. Moreover, the range of the diffraction scanning angle was from 10° to 80°, and the scanning rate used was 1°/min. As for the particle size analyses, a Malvern Panalytical company Zetasizer Ultra particle size analyzer system (Malvern Panalytical Ltd., Worcestershire, UK) was used. The sample used in the device had a volume of 5 µL.

### 2.3. Nanofluid Production

The nanofluid samples were fabricated at 0.001, 0.01, and 0.1 volume fraction (Φ). For calculating the Φ, the following equation was used:(1)$$\Phi =\left(\frac{V_{np}}{V_{np}+V_{bf}}\right)=\left[\frac{\left(\frac{m_{np}}{\rho_{np}}\right)}{\left(\frac{m_{np}}{\rho_{np}}\right)+V_{bf}}\right]$$
where, *V_np_*, *m_np_*, and *ρ_np_* are the volume, mass, and density of the silica nanoparticles, respectively, whereas the *V_bf_* is the volume of the basefluid. The *ρ_np_* was provided by the manufacturer as 2 g/cm^3^, and the *m_np_* were measured using a highly accurate ae-ADAM PW 214 analytical balance (Adam Equipment Ltd., Perth, Western Australia) with 10^−4^ g readability and ±2 × 10^−4^ g accuracy. Furthermore, the suspensions were then prepared through two stages. In the first stage, a magnetic stirrer device was used for 1 h to provide initial mixing between the nanopowder and the base fluid. Next, a Sonics Materials VCX 750 ultrasonic microprocessor, which is a probe type sonicator obtained from Fisher Scientific company. (Hampton, NH, USA), was used to further disperse the nanoparticles within the base fluid for 3 h. It is important to note that the ultrasonic device operated at 750 W net power output and 20 kHz with a 19 mm solid probe that is made of a high-grade titanium alloy (Ti-6Al-4V). Furthermore, the mixture temperature was controlled throughout the dispersion process using a BUENO BIOTECH company cooling and heating water bath (Nanjing, China), of type BGDC, with a 0.1 °C accuracy while being monitored using the ultrasonic temperature probe accessory, which is made of stainless steel and has a 1 °C accuracy. The previous suspension fabrication technique is widely used in the field and is known as the two-step nanofluid production method [30]. It is important to note that surfactants were not used in preparing the dispersions.

### 2.4. Stability Assessment of Nanofluids

One of the challenges that face nanofluids is to achieve a colloidally stable suspension. A stable suspension of nanofluids always shows better thermophysical properties than its unstable counterpart [31]. Hence, an assessment on nanofluid stability is very important before its application. The stability of nanofluids is assessed based on their concentration and aging [28]. Therefore, stability was studied for all freshly prepared and 20-day-old as-prepared samples. The stability was examined by a light absorbance test and zeta potential analysis. For the light absorbance test, it was performed through applying a digital colorimeter. The colorimeter displays the absorbance of visible light by a nanofluid sample that is proportional to its concentration [32]. After initial measurement, the samples were kept static for 20 days, and then the measurement was repeated. As for the zeta potential (ζ) analysis, it is regarded as one of the most reliable methods to examine the stability of nanofluids. Therefore, to validate the experimental results of a light absorbance test, zeta potential values were also estimated side by side to those of the other test. A zeta value describes the stability of nanofluids based on the electrophoretic mobility of dispersed nanoparticles [33]. A Malvern Zetasizer Nano was used to measure the zeta potential of the nanofluid samples that were initially prepared and after the 20-day period.

### 2.5. Boiling Pool Setup and Procedure

The boiling pool test was performed in a specially designed setup (Figure 1), which consisted of a rectangular closed chamber (length = 0.33 m, width = 0.30 m and height = 0.42 m) made of a 0.01 m thick SS sheet and placed on an iron stand. A coil heater of total 6 kW capacity is attached to the bottom surface of the chamber. Fireproof and heat resistance material and ceramic glass insulation are used to prevent heat leakage from the electric heater. An evaporator unit of a domestic freezer, which is mounted on the boiling chamber, works as a condenser to condense the vapor that comes out at the time of boiling. The entire boiling chamber was insulated with 0.015 m thick glass wool insulation to avoid heat leakage. The boiling chamber has an inlet and an outlet for the input and drainage of test fluid at the start and end of the experiment. A pressure gauge is mounted on the chamber to monitor the pressure inside the chamber during boiling. The test pool is also facilitated with a square-shaped glass window (made of heat-resistant quartz glass) of 0.0225 m^2^ to visualize the pool boiling process. The electrical power from the main supply unit comes to the heater through an AC variac. A voltmeter and an ammeter are connected across the circuit to display the voltage and current flow. Five K-type thermocouples (T_1_–T_5_) are connected to the heating surface (four thermocouples at the four corners and one at the center) to acquire the boiling surface temperature. The other two thermocouples (T_6_ andT_7_) are installed inside the boiling chamber to acquire bulk fluid temperature. The other ends of these thermocouples are connected to a temperature data logger. Before starting the experiment, the boiling chamber was cleaned with a water jet to remove dirt and other impurities. Then, the inside of the chamber was dried properly using a hand drier. After that, the roughness of the heating surface was measured by a Taylor Hobson Surtronic-25 portable roughness profiler. Next, 15 L of working fluid were injected through the inlet of the test pool. Subsequently, the heater was switched on and operated at 2 kW capacity. When the test fluid reached the saturation temperature, the boiling process was kept for 5 min to eradicate the bubbles dissolved in the fluid. The boiling of test fluid was performed under atmospheric conditions, and it is achieved by keeping the inlet open during experiment. The boiling temperatures of water and nanofluids were considered almost the same around 98.5 °C since the boiling point of nanofluids differ by 0.1–0.2 °C, which is within the error range of temperature measurement. After reaching the boiling point, the heat input to the system increased gradually by increasing the voltage input to the heater using the AC variac. The maximum power input to the heater was 5.7 kW. Therefore, the heater was operated just below its critical point to avoid burn out. Each sample was heated for 8–10 min to reach the maximum allowable heat flux. The corresponding voltmeter, ammeter, and temperature readings were saved and stored in a desktop computer to estimate PBHTC and CHF after the system reached a steady state of about 5 min. After the experiment, residual fluid was drained out, and surface roughness was measured. Next, the chamber was cleaned and dried to start a new experiment.

### 2.6. Experimental Conditions

The pool boiling performance of silica nanofluids was performed under saturated and nucleate boiling regions. To examine the effect of concentration and heat flux on PBHTC and CHF, an experiment was conducted with various volume fractions. The experimental conditions are presented in Table 1.

### 2.7. Surface Roughness and Contact Angle Measurements

The roughness of the boiling surface was measured using Taylor Hobson Surtronic-25 portable roughness profiler. The instrument has a specially engineered stylus that hovers on the test surface to measure its roughness. The instrument is battery-operated and portable, allowing it to be utilized freestanding on horizontal, vertical, or inverted surfaces, as well as bench mounted with fixturing for batch measurement and laboratory applications. The roughness was measured at several points on the heating surface, and the average was calculated. The roughness values of the uncoated and coated surface were measured before and after the boiling experiments with water and nanofluids. The contact angle between a test fluid and the uncoated stainless steel surface was measured using the sessile drop method [11]. The sessile drops of water and nanofluids on the boiling surface were analyzed and their contact angles were measured using a professional camera and imageJ [34], which is an open source image processing software.

### 2.8. Data Reduction and Uncertainty

The applied heat flux (*q*) to the heater surface is calculated knowing the electrical power and area of heating surface.
(2)$$q=\frac{P}{A}=\left(\frac{V\times I}{L\times W}\right)$$
where, *P* is the electrical power of the heater. *V* and *I* are the voltage and current input to the heater. *L* and *W* denote the length and width of the rectangular boiling chamber. The average heating surface temperature is calculated using the temperature acquired by the thermocouples *T*_1_–*T*_5_, as shown in Equation (3).
(3)$$T_{hs,avg}=\frac{\sum_{i=1}^{5}T_{i}}{5}$$

The bulk temperature (*T_b_*) of the pool of fluids is calculated as the average of the temperature attained by *T*_6_ and *T*_7_ thermocouples as follows:(4)$$T_{b}=\frac{T_{6}+T_{7}}{2}.$$

The PBHTC is calculated from applied heat flux, average heating surface temperature, and bulk temperature, as shown below.
(5)$$h=\frac{q}{\Delta T}=\frac{q}{\left(T_{hs,avg}-T_{s}\right)}$$
where, *h* denotes the PBHTC. ΔT represents the wall super heat temperature, which is the difference of average heating surface temperature, $T_{hs,avg}$, and saturation temperature, $T_{s}$. At boiling point, the bulk temperature (*T_b_*) is the same as saturation temperature (*T_s_*).

Experimental investigations are inherently associated with errors. Therefore, the measurement of any physical parameter should be repeated several times to test the repeatability of the experiment. In this experiment, the measurement of each physical parameter was repeated six times, and the result is presented as the mean of six repetitions± standard deviation. The uncertainty analysis of the measured parameters was performed according to the Kline and McClintock [35] as stated below:(6)$$U_{p}=\sqrt{\sum_{i=1}^{n}{\left(\frac{\partial P}{\partial x_{i}}\Delta x_{i}\right)}^{2}}$$
where, *U* is the uncertainty in the measured parameter *P* and *x* is the variables and Δx is the uncertainty in the instruments listed in Table 2.

The uncertainty in heat flux and PBHTC measurement were calculated using the following equations:(7)$$U_{q}=\sqrt{{\left(\frac{\partial q}{\partial V}\Delta V\right)}^{2}+{\left(\frac{\partial q}{\partial I}\Delta I\right)}^{2}+{\left(\frac{\partial q}{\partial L}\Delta L\right)}^{2}+{\left(\frac{\partial q}{\partial W}\Delta W\right)}^{2}}$$
(8)$$U_{h}=\sqrt{{\left(\frac{\partial h}{\partial q}\Delta q\right)}^{2}+{\left(\frac{\partial h}{\partial \left(T_{hs,avg}-T_{s}\right)}\Delta \left(T_{hs,avg}-T_{s}\right)\right)}^{2}}.$$

The maximum uncertainty in estimating PBHTC was found to be ±10.65%.

## 3. Result and Discussions

### 3.1. Feedstock Analysis

Figure 2 shows the FE-SEM micrographs, where it can be seen that the nanoparticles are nearly spherical in shape and are agglomerated. Furthermore, the EDS analysis (Figure 3) has shown that the sample consists of Si (35.4 wt %), O (56.4 wt %), and Ag (8.2 wt %) elements. The presence of Ag is due to the double-sided adhesive tape, and therefore, the reported purity of the nanopowder by the manufacturer can be assumed accurate (i.e., 99.5% purity). Moreover, the elemental distribution can be seen in Figure 4. As for the average particles size, it was found after analyzing the SEM micrographs with imageJ [36] software that they differ in size from 10.54 to 25.25 nm with a maximum occurrence of 17.24 ± 0.5 nm. Hence, the result is in good agreement with the provided specifications by the vendor.

On the other hand, it was found from the XRD analysis that the as-received sample was of cristobalite α-silica that had a tetragonal lattice structure. Figure 5 shows the XRD pattern that was obtained from the sample analysis and its corresponding (hkl) diffraction peak from the device database (PDF# 01-082-1233).

### 3.2. Nanofluids Physical Stability

The as-produced nanofluids samples, which were fabricated from dispersing 0.0001, 0.001, 0.01, and 0.1 volume fractions of the feedstock, are shown in Figure 6. These samples were analyzed in terms of their physical stability using both light absorbance and zeta potential analysis straight after their production and on the 20th day. The results of the absorbance test are presented in Figure 7. It is clear from Figure 7 that the initial absorbance values were within 0.512–0.567 for the samples of volume fraction from 0.0001 to 0.1. Then, the range of absorbance values change to values within 0.415–0.457 for the same volume fraction range after 20 days. The nanofluids with higher concentrations absorb more light than the lower concentrations. Hence, they show more absorbance values than dilute concentrations. Moreover, the absorbance values decreased with nanofluids aging, since smaller particles were only present in the dispersed phase, while bigger particles had already settled down [37]. The difference in absorbance values of nanofluids for a 20-day period is much less (shown by the right *Y*-axis), indicating minor change in the concentration of nanofluids. Hence, nanofluids can be considered stable. Moreover, the difference was seen to be larger in samples of higher concentrations, which indicates that increasing the nanoparticle loading has caused the suspensions to have lower physical stability.

On the other hand, the absolute zeta values of nanofluids for all concentrations and at different times are shown in Figure 8. The initial and final (i.e., after 20 days) zeta values are within the range of 51.5–48 mV. According to the previous studies that are available in the literature [38,39], a zeta value that is higher than 45 mV is considered stable. In the present case, the absolute zeta values even after 20 days remained above 45 mV. Moreover, the change in zeta values (shown by the right *Y*-axis) is also much less. Therefore, it can be concluded that the samples possessed good physical stability. This also confirms the experimental findings of the light absorbance test. Moreover, the zeta potential difference was more in samples with higher concentrations than those of lower concentrations. Hence, this demonstrates that nanofluid samples of higher concentrations are unstable, as shown previously with the light absorbance tests outcome.

### 3.3. Validity of Experiment

The validity of the pool boiling experimental setup is very essential to confirm the accuracy and reliability of the experimental outcomes. In order to achieve that, DI water was boiled in the test chamber, and the test result was compared with the well-known Rohsenow correlation [40,41] shown by Equation (9). Six test runs are executed with DI water, and the averaged data were compared with the Rohsenow correlation presented in Figure 9. The maximum and minimum absolute deviations were found to be 7.383% and 0.05% with 1.718% of average absolute (AAD). The experimental data reasonably fit well with the correlation data. Hence, the present experimental data are valid and reliable.
(9)$$\Delta T=\frac{h_{fg}}{C_{p,l}}C_{sf}{\left[\frac{q}{\mu_{l}h_{fg}}{\left(\frac{\sigma }{g\left(\rho_{l}-\rho_{v}\right)}\right)}^{\frac{1}{2}}\right]}^{\frac{1}{3}}{\mathrm{Pr}}^{n}$$
where *h_fg_* is the latent heat of vaporization. *C_p_* is the specific heat of fluid. *ρ* and *μ* are the density and viscosity of fluid. *σ* is the surface tension. Pr denotes Prandtl number. The subscripts *l* and *v* stand for liquid and vapor, respectively. *C_sf_* and *n* are constants whose values are *C_sf_* = 0.0132 and *n* = 1 for polished steel surface and pure water.

### 3.4. Boiling Characteristics Curves

Pool boiling characteristics curves of silica nanofluids are presented in Figure 10. In Figure 10, the heat flux is compared with the wall superheat temperature. The total graph area is divided into two regions: natural convective region and nucleate boiling region. In the natural convective region, the heat transfer rate is small, and its values differ closely for water as well as nanofluids. Therefore, in this regime, the slopes of curves are small, and they are very close to each other. However, at the nucleate boiling region, the heat transfer is more, and for this, the slopes are higher; even the heat transfer values varies widely for all test fluids. The curves shifted to the left side of the boiling curve of water due to nanoparticle addition in it. The shifting of curves signifies that nanoparticle addition leads to an increase in heat transfer for identical superheat temperature [42,43]. Therefore, the heater performance increased. The shifting continued with 0.0001 and 0.001 volume fractions, but it turns toward the right with the two other concentrations (0.01 and 0.1 volume fractions). As found earlier, that sedimentation is more in nanofluids of higher concentrations. Hence, there is more nanoparticle deposition on the boiling surface with higher concentrations. Depositing nanoparticles creates a layer that hinders the heat transfer from the boiling surface to working fluids. Therefore, the heat transfer rate in nanofluids decreased with higher concentrations accompanied by a rise in the wall superheat temperature [44]. The operational safety of the boiling equipment depends on the wall superheat. The improvement of heat transfer at lower wall superheat temperature presents the advantage of adding nanoparticles in base fluids. However, the addition of nanoparticles needs to be controlled to avoid heat transfer deterioration.

### 3.5. PBHTC and Its Enhancement

The PBHTC values of nanofluids are presented in Figure 11. In Figure 11a, the PBHTC is presented as the function of wall superheat temperature, and the same is plotted against heat flux in Figure 11b. By observing both figures, it is evident that PBHTC significantly increased with nanoparticles addition in water. The enhancement signifies several things: first, thermal conductivity enhancement in nanofluids, which boosts heat transfer from the heating surface. Second, there are improvements in particle mobility and Brownian motion due to bubble movement during boiling. This can improve the heat transfer at the interface of the heating surface and bulk fluid. Third, there is the development of a porous layer of nanoparticle disposition on the heating surface. The deposited nanoparticles facilitate in the nucleation site growth over the boiling surface, and consequently, the heat transfer rate enhances. The PBHTC toward the natural convection side does not show distinctive values for nanofluids with all concentrations and water, since the transferred heat flux is less, and its values are near each other. However, the PBHTC values at the nucleate boiling region expanded widely since the heat transfer rate is more at that region. The PBHTC increases with 0.0001 and 0.001 volume fractions, and it descends with 0.01 and 0.1 volume fractions. Such decrement is due to the additional particle deposition on the heater surface, preventing heat transfer from the surface to bulk fluid. The present trend of PBHTC is similar to the observation reported by Golkar et al. [45].

The enhancement in PBHTC as compared to water is expressed here as the PBHTC ratio $\left(h_{r}\right)$, which is defined below:(10)$$h_{r}=\left(\frac{h_{nf}}{h_{bf}}\right).$$

The PBHTC ratio values for different concentrations are presented in Figure 12. In Figure 12, it is clear that all PBHTC ratios are greater than unity, which is the base value of the PBHTC ratio for water. Hence, the enhancement was achieved due to the dispersed nanoparticles within the base fluid. However, the increase in the level of enhancement was possible up to a certain concentration, after which it started to decrease. The maximum PBHTC value is 1.33, which belongs to 0.001 volume concentration. The lowest enhancement is at the highest concentration (i.e., 0.1 volume fraction) with a value of 1.07.

### 3.6. CHF and its Enhancement

The maximum points on Figure 13 are *CHF* values of test fluids where heat transfer reaches its maximum value, above which the heating system may burn out and might affect the operational safety. The voltage and current reached their highest permissible values, and the coil heater became red hot due to excessive heating. At *CHF*, the power input changed significantly, and at a small wall superheat temperature, the heat transfer abruptly increased. The *CHF* enhancement for silica nanofluids as compared to water is presented here as critical heat flux ratio (*CHF_r_*), which is defined as follows:(11)$$CHF_{r}=\left(\frac{CHF_{nf}}{CHF_{bf}}\right).$$

The *CHF* and *CHF_r_* values of water and nanofluids at studied concentrations are presented in Figure 13. The *CHF* value of water increased with nanoparticle addition up to 0.001 volume fraction. However, further addition of nanoparticles (0.01 and 0.1 volume fractions) reduces the *CHF*. The thermal properties of base fluids are enhanced by nanoparticle dispersion in it, and because of that, the thermal transport capacity of base fluids radically improves. Hence, the nanofluids can encounter high heat flux compared to their base fluids, which cannot. Moreover, the nanoparticle deposition generates additional nucleation sites, resulting in heat transfer improvement from the boiling surface. However, at higher concentrations, the deposition is more, and the deposited layer is thicker. The thicker layer hinders heat transfer by increasing thermal resistance. The *CHF_r_* values of nanofluids are all greater than unity i.e., above the baseline (with water). Hence, CHF enhancement is possible with nanofluids. The trend of experimental CHF enhancement is similar to the experimental works reported by Zafar et al. [46] and Alam et al. [47]. The maximum value of the *CHF_r_* ratio is 1.33, and the minimum is 1.07 with 0.001 and 0.1 volume fractions.

### 3.7. Wettability, Surface Contact Angle, and Capillarity

Pool boiling depends on the wettability of the heating surface. Surface contact angle is the measure of wettability of test fluids. The contact angles between the heating surface and the test fluids are described in the Figure 14. The contact angle decreased with increasing concentration of nanofluids. The nanoparticles’ deposition altered the surface characteristics and enhanced the wettability of the surface. As a result, the contact angle decreased. The silica nanoparticles, due to their bigger size, thermophysical properties, and higher density as well as mass compared with the liquid molecules, squeeze the droplet toward the solid surface, resulting in a smaller contact angle [48,49]. The surface contact angle depends on several factors such as surface roughness [50], hydrophilicity or hydrophobicity [23], adhesion force [51], surface tension [52,53], and capillarity [54]. The decrease in contact angle may be attributed to the decrease in liquid surface tension due to an increase in interfacial tension between liquid–solid interfaces. The increase in interfacial tension provides a tendency to attach to the boiling surface, and liquid films form regularly, enhancing boiling heat transfer and CHF.

The PBHTC and CHF enhancements are also explained by capillarity of the porous layer formation due to nanoparticle deposition on the heating surface. The capillarity is related to the contact angle in the following relation:(12)$$L_{C}=\frac{2\sigma }{gR\rho_{l}}\cos\theta .$$

In Equation (12), σ is the surface tension, g represents acceleration due to gravity, *R* is the capillary radius, $\rho_{l}$ denotes the density of liquid, and θ represents the contact angle.

Therefore, a decrease in contact angle also enhances the capillary action in the porous layer. The capillary action on the heater surface greatly influences the nucleation, bubble dynamics, and bubble departure frequency, resulting in heat transfer enhancement. Moreover, the tapped nanofluids in the porous structure moves a greater distance, and therefore wet the boiling surface. This resulted in a delay in the dry out and cooling of the hotspot, and hence enhancing the CHF. However, according to Sarafraz et al. [54], the capillarity alone does not explain the mechanism for pool boiling and CHF enhancements in nanofluids. The porosity of the deposited layer at lower concentrations remains prominent, but the high deposition rate of nanoparticles with higher concentrations makes the porous layer impassable to develop capillary action. As a result, the CHF performance diminished. Therefore, particle concentration and size optimization are important to reduce sedimentation on the heater surface.

### 3.8. Surface Roughness

Figure 15 presents the roughness of the boiling surface with the changing volume fraction of nanofluids. The roughness decreases with the increase in concentration. The particle deposition on the boiling surface increases with the volume fraction due to a greater number of particles present in the fluid. The roughness of the boiling surface can be increased by particle deposition if the particle size is bigger than the initial roughness of the heating surface [55]. However, in the present experiment, the particle size is in nano scale and the roughness is in micron scale. Therefore, opposite phenomena were observed in this study. The pool boiling and CHF critically depend on the roughness of the heating surface. The increase in surface roughness aids in pool boiling heat transfer performance by providing more nucleation sites, overheating the surface. More development in the nucleation site is observed because of the porous layer formation over the boiling surface with the dispersion of silica nanoparticles in base fluids. However, excess particle deposition cause thickening of the deposited layer, which leads to an increase in thermal resistance, and hence, the heat transfer decreases. This hypothesis can be well explained with the present experimental observations. With lower concentrations (0.0001 and 0.001 volume fractions), nanofluids showed limited particle deposition, producing more nucleation sites that resulted in heat transfer enhancement. In addition, with higher concentrations (0.01 and 0.1 volume fractions), particle deposition was more; developing a thick layer that further impeded heat transfer. The result reflected an increase in PBHTC and CHF for 0.0001 and 0.001 volume fractions and decrement with 0.01 and 0.1 volume fractions. According to Kim et al. [56], the deposited layer of nanoparticles produces a continuous change in surface topology (geometry), which affects the boiling heat transfer in nanofluids. The change in the surface geometry greatly depends on the nanoparticle concentration. In the case of nanofluids with low particle concentration (0.0001 and 0.001 volume fractions), there is a slight change in the heating surface. However, when the concentration rises, the surface deposition of the nanoparticles thickens, and more micro-sized structures develop on the heated surface due to silica nanoparticle aggregation.

### 3.9. Bubble Frequency and Bubble Departure Diameter

The nucleation site density, bubble departure diameter, and bubble departure frequency all influence the surface superheat temperature or heat transfer coefficient. The bubble departure diameter is determined by surface tension and buoyancy forces applying Fritz correlation [57]:(13)$$D_{bd}=20.8\times 10^{-3}\sqrt{\frac{\sigma }{g(\rho_{l}-\rho_{g})}}\theta .$$

Therefore, theoretically, the departure happens when buoyancy overcomes the inertia, drag force, and surface tension, which prevent the bubbles from detaching.

Phan et al. [58] developed a model that takes into account the Fritz correlation [57] as well as the influence of the energy factor as a contribution from the wetting effects:(14)$$D_{bd}=626.977\times 10^{-3}\sqrt{\frac{\sigma }{g(\rho_{l}-\rho_{g})}}f\left(\theta \right)$$
(15)$$f\left(\theta \right)=\frac{1}{2}+\frac{3}{4}\cos\theta -\frac{1}{4}\cos^{3}\theta$$
where, $f\left(\theta \right)$ is the volume ratio of a truncated sphere to a whole sphere of the same diameter. Therefore, the bubble diameter increases as the surface wettability increases, as shown in Equation (13).

McFadden and Grassmann [59] presented a mathematical relationship between bubble diameter and bubble frequency:(16)$$f_{B}=\lambda {\left(\frac{g}{D_{bd}}\right)}^{0.5}$$
where, λ is the proportionality constant. Ivey [60] suggested λ = 0.9 when *D_bd_* > 0.3 cm but when *D_bd_* < 0.3 $f_{B}$ should be scaled by $D_{bd}$^−2^ instead of $D_{bd}$^0.5^.

In the present experiment, the bubble diameter and bubble frequency are presented as the function of nanofluids volume fraction in Figure 16. The bubble diameter increased up to 4.48% with the increasing volume fraction because the surface wettability increased. Dong et al. [61] reported that the bubble diameter is bigger with smooth surfaces. This might be related to the fusion of bubbles on the smoother surface, which developed during the boiling of nanofluids with higher volume fractions. The increase in bubble diameter causes a decrease in heat transfer, since the larger bubbles stick to the surface and act as insulation to the heat transfer from the heating surface. Thus, the boiling heat transfer performance decreased with high volume fractions. The bubble frequency slightly decreased with increasing volume fraction. The bubble departure frequency decreased from 75.85 per second in case of water to 74.21 with 0.1 volume fraction, which indicates a 2.17% increase in departure frequency. The decrease in bubble departure frequency can be attributed to the increase in bubble volume due to the increase in bubble diameter. The bigger bubbles take a longer time to descend from the boiling surface due to their inertia, resulting in a decrease in bubble departure frequency.

### 3.10. Comparison with Published Results

The PBHTC and CHF are compared with the published results. Similar kinds of studies from the literature were compared with the present experimental work for better understanding of the pool boiling mechanism. The pool boiling heat transfer entirely depends on the bulk effect and surface effect of nanofluids. The bulk effect depends on nanofluids’ thermophysical properties and the surface effect on the boiling surface morphology, nanoparticle deposition, as well as solid–fluid and water interactions. The interactions among these parameters are very complex and do not indicate the best nanofluids for pool boiling applications. However, in order to evaluate the boiling performance of silica nanofluids over a flat stainless steel surface, the PBHTC ratio and CHF ratio are compared with the published literature. The PBHTC ratio is compared with the experimental results of Kamel et al. [62,63] and Abdollahi et al. [64] The CHF ratio is compared with the same predicted by the Kandikar model [65] and Zuber models [66], and furthermore, it is compared with the experimental data presented by Hu et al. [42] in Figure 17. In Figure 17a, the PBHTC ratio is better than the same obtained by Kamel et al. [62,63], but it is lower than the PBHTC ratio reported by Abdollahi et al. [64]. There are several reasons to cite behind such occurrences. First, the volume concentration range used by Kamel et al. [62,63] is much less than that in our study, although they used nanoparticles i.e., MgO and hybrid Al_2_O_3_-CeO_2_, which have much higher thermal conductivities than silica nanoparticles. In addition, there might be a deposition of nanoparticles on the heater surface, hindering heat transfer. Hence, the highest PBHTC ratio enhancement was restricted to 1.22 and 1.25 in their studies, while in the present study, PBHTC reached up to 1.33 at 0.001 volume fraction. Again, Abdollahi et al. [64] used Fe_3_O_4_ with concentrations that have a similar range to those used in the representative experiment. However, they have used an external magnetic field, which enhances the PBHTC in nanofluids. Hence, PBHTC is highest in their case with 1.43 at 0.001 volume fraction. In Figure 17b, the Kandikar [65] and Zuber [66] models under predict the CHF enhancement and do not account for the effect of particle concentration on the CHF ratio. Hence, it remained very close to the baseline and showed small changes with concentration rise. The highest CHF ratios are 1.07 with the Kandikar model [65] and 1.02 with the Zuber [66] model. Thus, it shows the limitations of the prevailing theoretical models in predicting CHF enhancement with nanofluids. Then, the experimental CHF ratio data were compared with the experimental data of Hu et al. [42]. The CHF ratios obtained by Hu et al. [42] are higher than the same in the present experiment. Hu et al. [42] studied SiO_2_ boiling over platinum heater wire, but here, flat stainless steel is used. The experimental condition was different from the present experiment. However, the trend of CHF enhancement is quite similar. The highest CHF ratio in the experiment of Hu et al. [42] was 1.69, which indicates almost 70% enhancement, and it is 1.33 in the present experiment, which indicates 33% enhancement in CHF as compared to water. Finally, the addition of silica nanoparticles in water has the potential to improve the pool boiling performance and CHF enhancement with proper heating surface, orientation, and surface topology together with optimum nanoparticle concentration and surface roughness.

## 4. Conclusions

The pool boiling heat transfer performance of water-based silica nanofluids was studied experimentally. Silica nanofluids with various concentrations were prepared, and their stability was studied as well as their boiling characteristics on a flat stainless steel surface. The PBHTC and CHF performances of nanofluids were estimated and compared with water. The surface contact angle, wettability, and roughness were examined over different concentrations of dispersed silica particles. Finally, the bubble departure diameter and bubble departure frequency over the studied concentration range were estimated. The main findings of the present study can be summarized as follows:Light absorbance and zeta potential analysis verified the stability of nanofluids. The nanofluids remained stable for more than 20 days.The PBHTC and CHF were enhanced with lower concentrations; however, they decreased with higher concentrations. A maximum of 1.33 times enhancements greater than water in PBHTC and CHF was obtained in nanofluids with 0.001 volume fraction. The deposition of nanoparticles over the heater surface is the main reason of heat transfer deterioration at a higher level of nanofluids concentration.The wettability and capillarity of nanofluids increased with the decrease in surface contact angle. The increases in wettability and capillarity enhance pool boiling heat transfer and CHF in nanofluids.Surface roughness decreased with particle inclusion due to particle deposition on the heating surface. The reduction is more pronounced with higher volume fractions; hence, the pool boiling performance reduced.The bubble departure diameter of nanofluids was enlarged with volume fraction as the wettability enhanced and due to the bubble merging on the smooth surface. The bubble departure frequency slightly decreased with rising concentration due to an increase in bubble size. Such occurrences influence the pool boiling performance of nanofluids.

It is seen that several factors such as nanoparticle concentration and surface roughness influence pool boiling enhancement. Hence, the authors recommend an optimization operation to determine the ideal concentration with which maximum enhancement in pool boiling can be achieved. A study on the reusability of formerly boiled nanofluids and its stability can be a good study for practical implication of nanofluids boiling. Scanning electron microscopy of the pre and post boiling surface will be appropriate for better understanding the boiling surface topology. Moreover, studies on the contact angles and wettability of pre and post boiling surfaces will be helpful to understand the surface wettability effect on boiling. Finally, a visualization study is recommended for better interpretation of the bubble formation, bubble diameter, and bubble frequency in pool boiling heat transfer of silica nanofluids.

## Figures and Tables

**Figure 1 nanomaterials-11-02138-f001:**
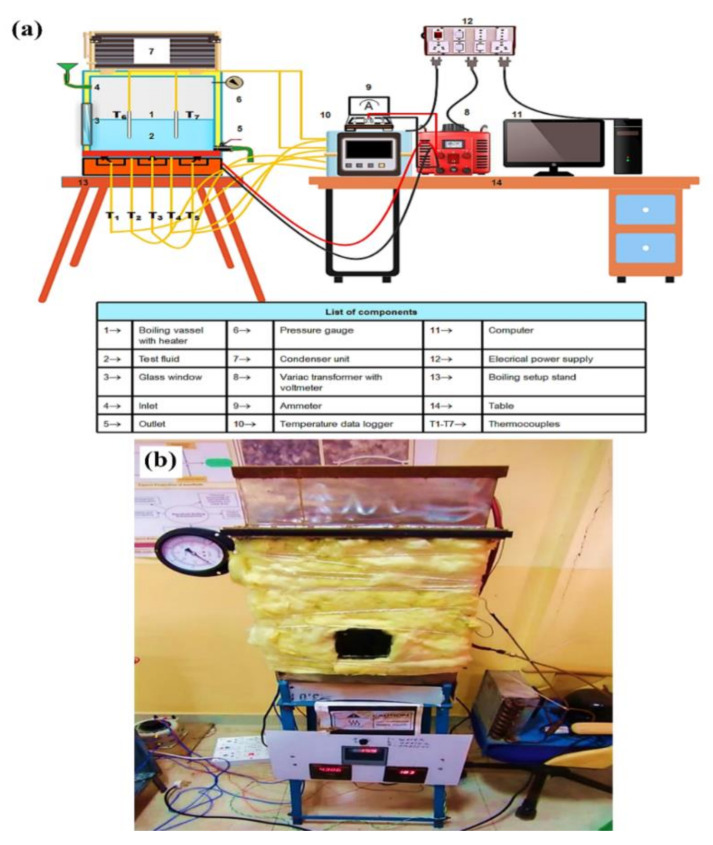
Experimental test rig, where (**a**) demonstrates the schematics design and (**b**) illustrates the actual system.

**Figure 2 nanomaterials-11-02138-f002:**
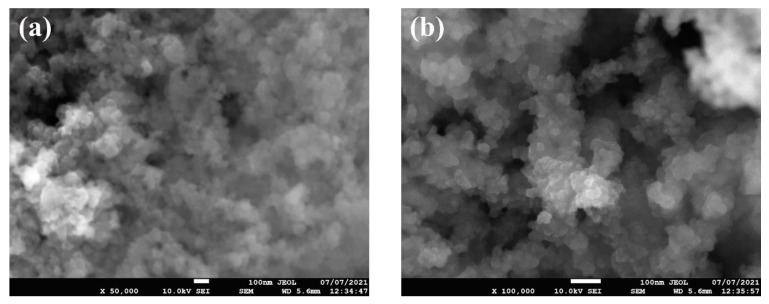
SEM micrographs of the as-received powder, where (**a**) shows the low-magnification image and (**b**) demonstrate the high-magnification image.

**Figure 3 nanomaterials-11-02138-f003:**
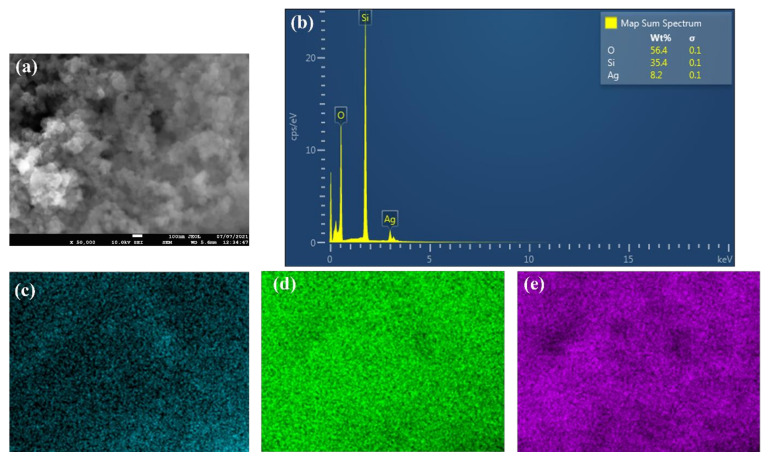
EDS analysis of the as-received powder, where (**a**) shows the SEM image in which the analysis was conducted, (**b**) demonstrates the elements’ wt %, and (**c**–**e**) illustrate the distribution of Ag, Si, and O elements, respectively.

**Figure 4 nanomaterials-11-02138-f004:**
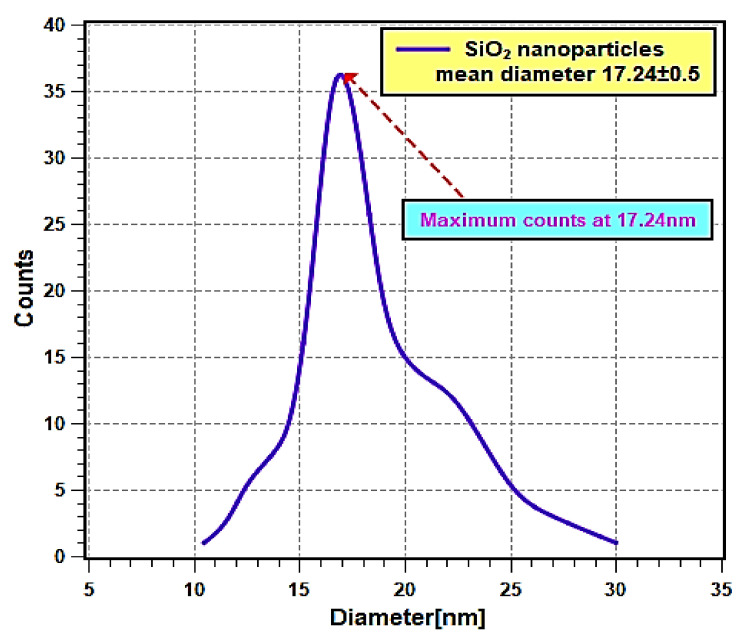
Nanoparticles mean diameter distribution curve.

**Figure 5 nanomaterials-11-02138-f005:**
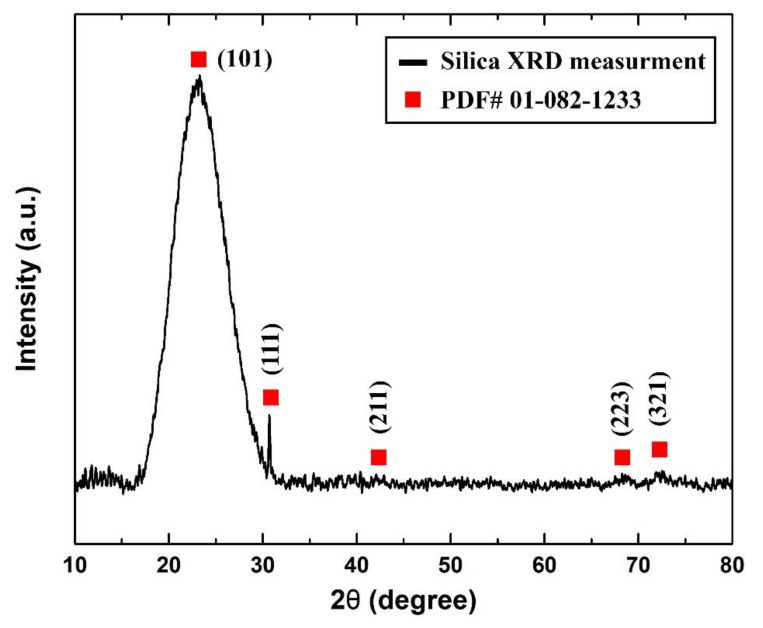
As-received sample measured XRD pattern.

**Figure 6 nanomaterials-11-02138-f006:**
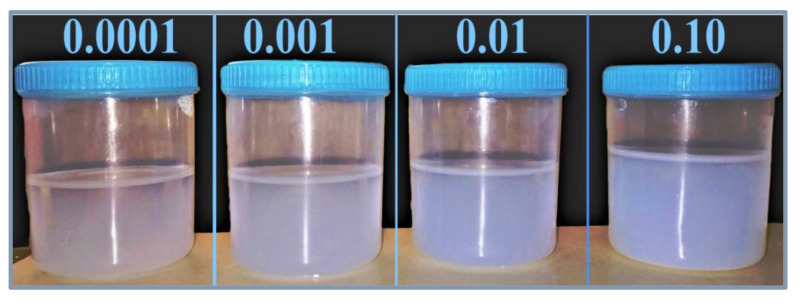
Images of the as-prepared SiO_2_–water nanofluid samples.

**Figure 7 nanomaterials-11-02138-f007:**
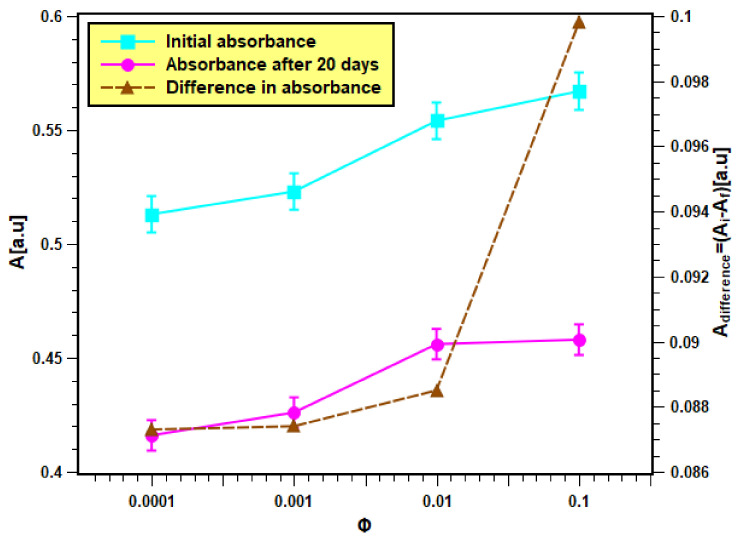
Light absorbance result of the as-prepared nanofluid samples.

**Figure 8 nanomaterials-11-02138-f008:**
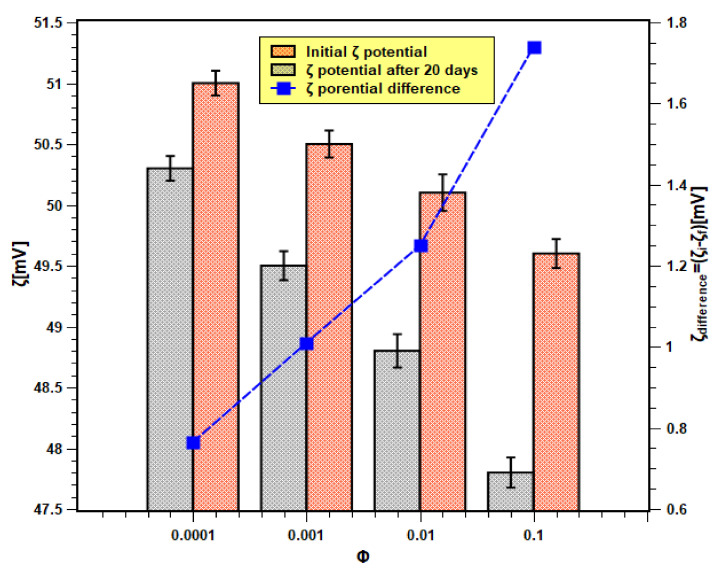
Zeta potential analysis results of nanofluids.

**Figure 9 nanomaterials-11-02138-f009:**
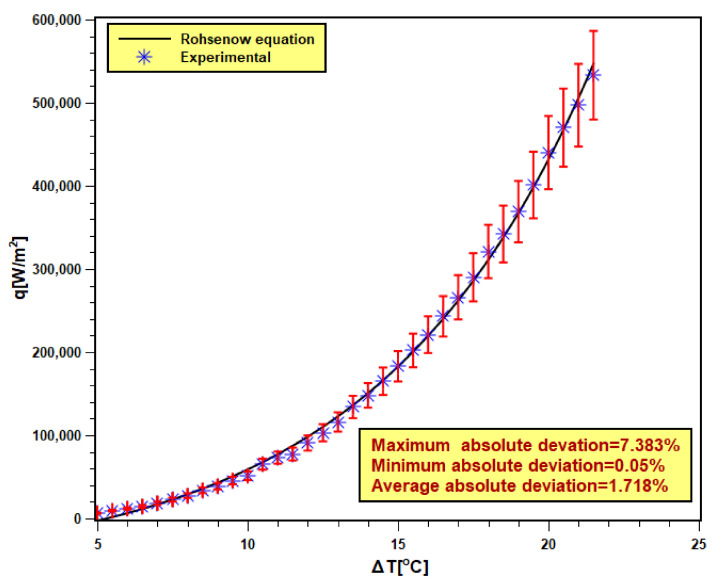
Validation of experimental data with Rohsenow equation [40,41].

**Figure 10 nanomaterials-11-02138-f010:**
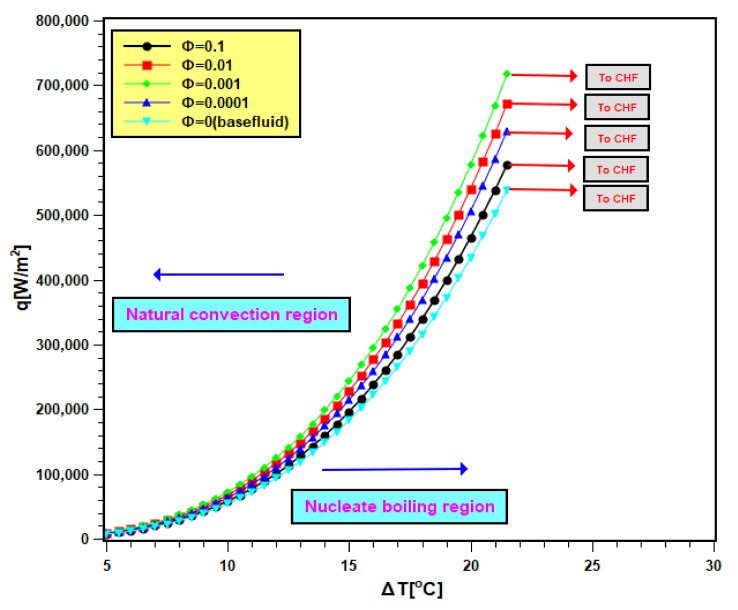
Boiling heat transfer characteristics curve of nanofluid.

**Figure 11 nanomaterials-11-02138-f011:**
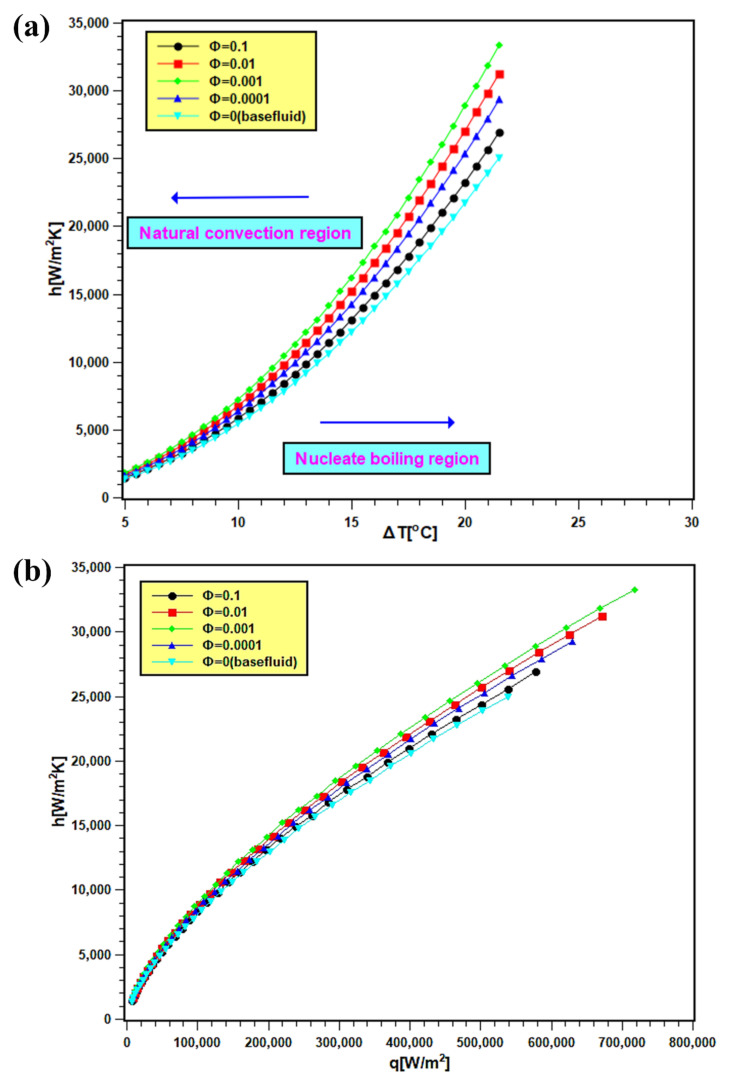
PBHTC of silica nanofluids, where (**a**) is against the wall super heat temperature and (**b**) is against the heat flux at different volume fractions.

**Figure 12 nanomaterials-11-02138-f012:**
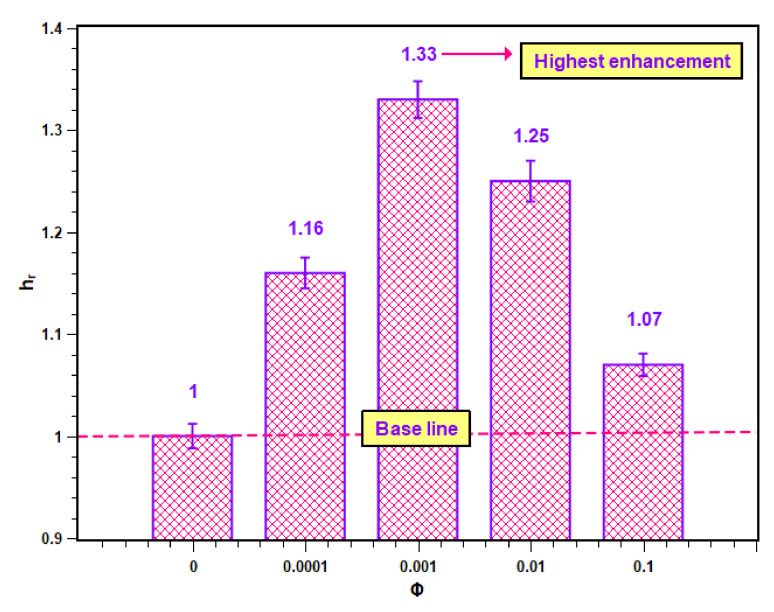
Enhancement in heat transfer coefficient of SiO_2_–water nanofluid.

**Figure 13 nanomaterials-11-02138-f013:**
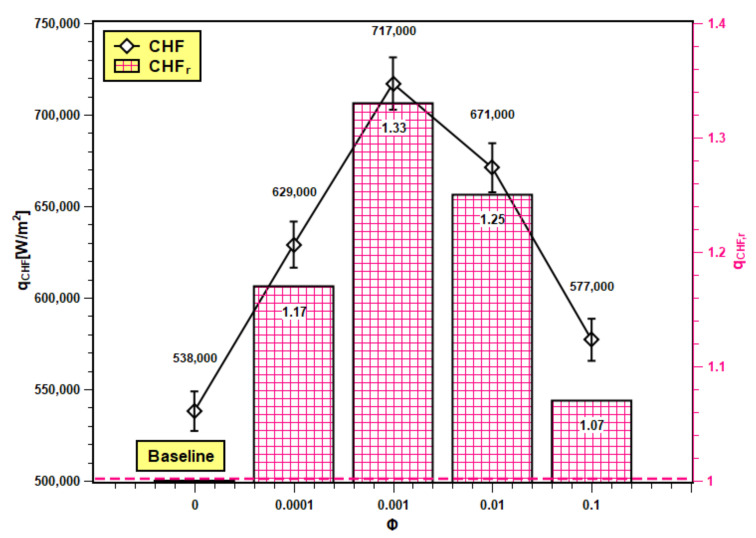
CHF and its enhancement in SiO_2_ nanofluids.

**Figure 14 nanomaterials-11-02138-f014:**
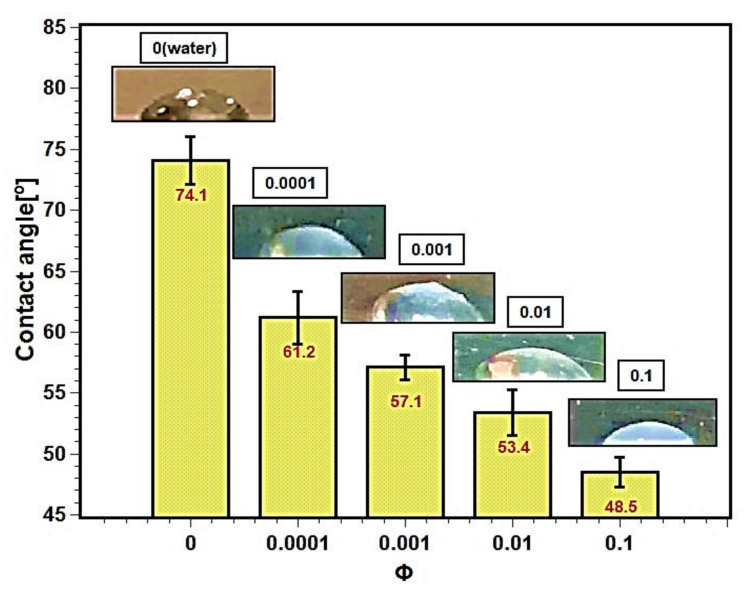
Surface contact angles for silica–water nanofluids at different concentrations.

**Figure 15 nanomaterials-11-02138-f015:**
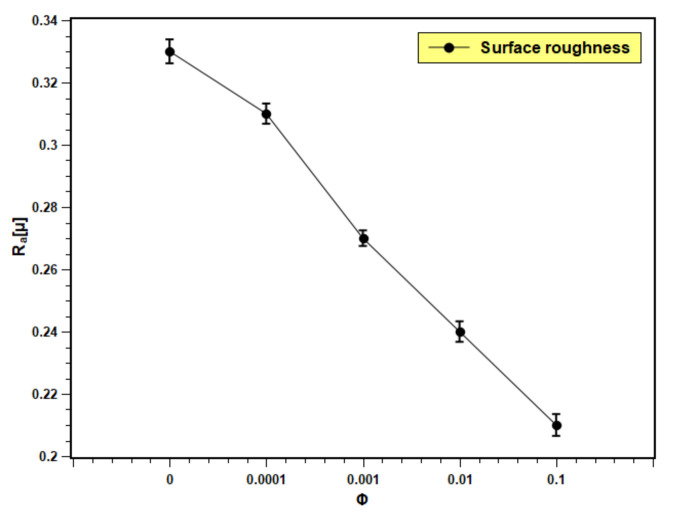
Boiling surface roughness profile with test fluids at different concentrations.

**Figure 16 nanomaterials-11-02138-f016:**
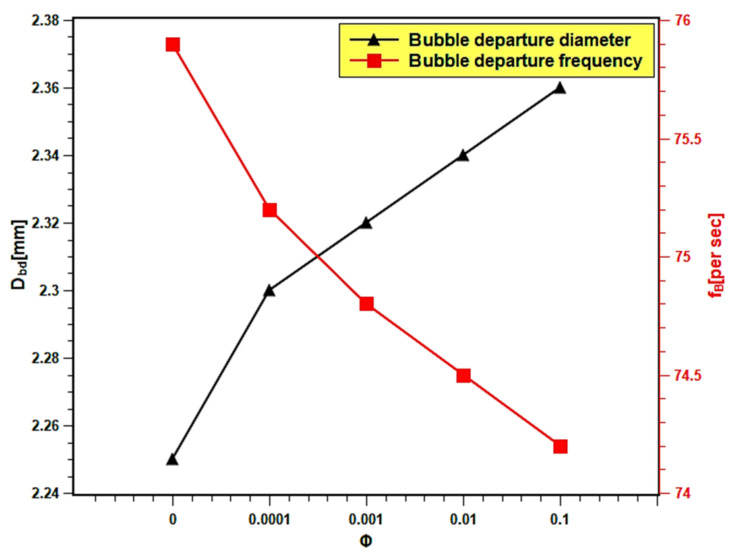
Bubble departure diameter and bubble frequency variation with concentration.

**Figure 17 nanomaterials-11-02138-f017:**
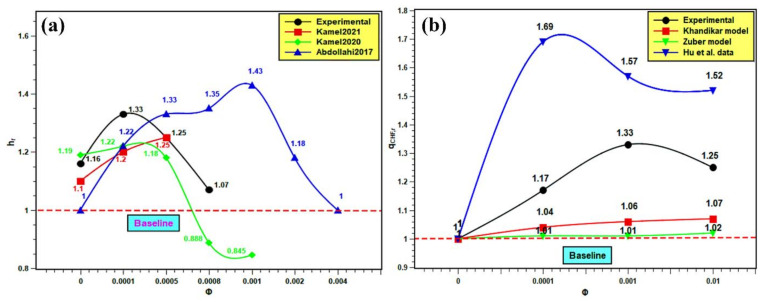
Comparison with published literature: (**a**) for PBHTC enhancement and (**b**) for CHF enhancement.

**Table 1 nanomaterials-11-02138-t001:** Experimental conditions applied to the present study.

Parameter	Range
Particle size range	10.55–25.25 nm
Volume fraction	0.0001–0.1
Heat flux input	6.7–577.483 kW/m^2^
Boiling temperature	97.5–101.12 °C
Boiling region	Nucleate boiling region

**Table 2 nanomaterials-11-02138-t002:** Uncertainty of the instruments.

Instruments	Parameters	Uncertainty
Thermocouple	Temperature	±0.1 °C
Voltmeter	Voltage	±0.3 V
Ammeter	Current	±0.01 A
Pressure gauge	Pressure	±0.05
Chino KR2000 data logger	Temperature	±0.1%
Taylor Hobson Surtronic-25	Surface roughness	±0.015

## Data Availability

All data are available upon request from any of the authors.

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
