# Peer review of "Pool Boiling Amelioration by Aqueous Dispersion of Silica Nanoparticles"

_nanomaterials, 2021, doi:10.3390/nano11082138_

Round 1

Reviewer 1 Report

Comments on “Pool Boiling Amelioration by Aqueous Dispersion of Silica Nanoparticles”

  • Check the English and ensure typos and grammatical errors are addressed in the next version of the paper.
  • Why silica nanoparticles were selected for the present study? What is the main driver for using silica despite its instability as nanofluid?
  • Introduction needs to be further enriched by reviewing state-of-the-art studies on boiling of nanofluids and their thermal performance assessment aiming at highlight the importance of further research in this area. For example, searching the literature, following papers are suggested to be read and used: Thermal analysis of a binary base fluid in pool boiling system of glycol–water alumina nano-suspension. Boiling flow of graphene nanoplatelets nano-suspension on a small copper disk.  Boiling heat transfer characteristics of graphene oxide nanoplatelets nano-suspensions of water-perfluorohexane (C6F14) and water-n-pentane.  Pool boiling heat transfer to CuO-H2O nanofluid on finned surfaces.  Transient pool boiling and particulate deposition of copper oxide nano-suspensions.  Experimental study of the effect of various surfactants on surface sediment and pool boiling heat transfer coefficient of silica/DI water nano-fluid.  Providing a model for Csf according to pool boiling convection heat transfer of water/ferrous oxide nanofluid using sensitivity analysis. 
  • Can authors add a scanning electron microscopy analysis from the boiling surface discussing the roughness and irregularity of the surface?
  • How stable were the nanofluids before and after the experiments?
  • How authors measured 0.0001 fraction of nanoparticles and how they dispersed such a small fraction inside base fluid?

All in all, the paper can be accepted once above comments are addressed.

Author Response

The authors are very much thankful to the respected reviewer for accepting to review our manuscript and sharing his valuable time. We cordially acknowledge the useful comments and recommendations made by the respected reviewer on our manuscript. We have tried to revise the manuscript accordingly and the detailed corrections are listed below point by point. The respected reviewer can find all the modifications and corrections in the revised manuscript written in red text. The authors respond to the respected reviewer queries are as follows:

Comment-1: Check the English and ensure typos and grammatical errors are addressed in the next version of the paper.

Response: The authors cordially acknowledge the valuable comments and suggestions provided by the respected reviewer for the betterment of the present manuscript. The authors have carefully checked the English language, grammatical errors, and typos presented in the manuscript. Thank you very much.

Comment-2: Why silica nanoparticles were selected for the present study? What is the main driver for using silica despite its instability as nanofluid?

Response: The authors are very much thankful for this question. The reason behind selecting silica nanoparticles is because this type of nanomaterial is cheaper than other nanoparticles. Also, there are relatively few studies that intensively focuses on this type of dispersions:

10.1016/j.proeng.2015.05.030

10.1016/j.icheatmasstransfer.2015.01.002

10.1016/j.ijthermalsci.2015.07.008

10.1007/s11051-014-2564-2

The previous has encouraged us to further investigate this type of nanofluids.

Comment-3: Introduction needs to be further enriched by reviewing state-of-the-art studies on boiling of nanofluids and their thermal performance assessment aiming at highlight the importance of further research in this area. For example, searching the literature, following papers are suggested to be read and used: Thermal analysis of a binary base fluid in pool boiling system of glycol–water alumina nano-suspension. Boiling flow of graphene nanoplatelets nano-suspension on a small copper disk. Boiling heat transfer characteristics of graphene oxide nanoplatelets nano-suspensions of water-perfluorohexane (C6F14) and water-n-pentane. Pool boiling heat transfer to CuO-H2O nanofluid on finned surfaces. Transient pool boiling and particulate deposition of copper oxide nano-suspensions. Experimental study of the effect of various surfactants on surface sediment and pool boiling heat transfer coefficient of silica/DI water nano-fluid. Providing a model for Csf according to pool boiling convection heat transfer of water/ferrous oxide nanofluid using sensitivity analysis.

Response: The authors cordially acknowledge the suggestions came from the respected reviewer regarding the betterment of the introduction portion of the manuscript. As per the suggestions the authors have included all the suggested literature in the revised version of the manuscript and improved the introduction section. Thank you very much.

Comment-4: Can authors add a scanning electron microscopy analysis from the boiling surface discussing the roughness and irregularity of the surface?

Response: The authors like to apologies to the respected reviewer and state that they cannot provide the scanning electron microscopy of the boiling surface at this time. This is because the required facility is not currently available at their institutes. However, they have included the respected reviewer recommendation as part of the future scope of the present work. Please see the last paragraph in the Conclusions Section. We apologies again for any inconvenience. Thank you very much.

Comment-5: How stable were the nanofluids before and after the experiments?

Response: The nanofluids are less stable after the boiling heat transfer since some volume of the basefluid got evaporated while leaving the solid particles in the remaining quantity, and hence the remaining samples become more concentrated with nanoparticles and relatively instable. The authors consider this topic as the future scope of the present work to be later on studied in a more intensive manner. Please see the last paragraph in the Conclusions Section.

Comment-6: How authors measured 0.0001 fraction of nanoparticles and how they dispersed such a small fraction inside base fluid?

Response: We thank the respected reviewer. Kindly note that the authors used a very precise and highly accurate weighing balance machine of ±2 × 10-4 g accuracy to measure the weight of nanoparticles. Moreover, the authors have used 15 liters of basefluid as their sample before injecting it in the form of nanofluid into the test pool. For preparing nanofluids of such volume requires good amount of nanoparticles to be dispersed in the basefluid. Hence, the 0.0001 particle concentration with 15 liters of basefluid is actually not hard to achieve although it seems to be very small quantity.

Comment-7: All in all, the paper can be accepted once above comments are addressed.

Response: The authors highly appreciate the respected reviewer words and thank him very much.

Reviewer 2 Report

Pool Boiling Amelioration by Aqueous Dispersion of Silica Nanoparticles

The present paper investigated the pool boiling heat transfer characteristics and performance of water-based silica nanofluids under the nucleate boiling region. The PBHT and CHF were analyzed under different particle concentrations. Effects of surface wettability, contact angle, surface roughness on heat transfer performance were investigated.

Considering some innovations of this work, it can be considered for publication if the article is modified satisfactorily. The following suggestions are provided for further improving the quality of the manuscript.

  1. The abstract needs to be modified. The primary conclusions and innovation should be supplemented.
  2. The introduction structure needs to be modified. The third paragraph is too long and needs to be broken up according to the research categories.
  3. The major work and innovation need to be supplemented at the end of the introduction.
  4. The symbols in text and formulas need to be unified. The font and format of the formulas need to be checked and modified. Authors should check the whole paper more carefully.
  5. In the results part, the authors just showed the phenomenon of the experiments. The reason needs to be analyzed either. For example, in Boiling characteristics curves, there is no analysis of reason. The authors should explain the reason why the boiling curves of 0.01 and 0.1 turn towards the right.
  6. Figure 13 shows the contact angles at different concentrations. The contact angles seem to be measured before boiling. The contact angles measurement after the boiling should be supplemented. Otherwise, the effect of wettability can not be confirmed. Generally, the hydrophilic surface shows better boiling performance.
  7. There is an error in figure 14. The value of the x-axis is wrong. Authors should check the whole paper carefully.
  8. The font and format errors are found on page 21.
  9. The pictures of the bubbles need to be supplemented. The difference in bubble departure diameter needs to be explained.

Author Response

The authors are very much thankful to the respected reviewer for accepting to review our manuscript and sharing his valuable time. We cordially acknowledge the useful comments and recommendations made by the respected reviewer on our manuscript. We have tried to revise the manuscript accordingly and the detailed corrections are listed below point by point. The respected reviewer can find all the modifications and corrections in the revised manuscript written in red text. The authors respond to the respected reviewer queries are as follows:

Comment-1: The abstract needs to be modified. The primary conclusions and innovation should be supplemented.

Response: The authors are thankful to the respected reviewer for his comments and suggestions. The abstract has been revised by supplementing primary conclusions and innovation, as advised. Thank you very much.

Comment-2: The introduction structure needs to be modified. The third paragraph is too long and needs to be broken up according to the research categories.

Response: The structure of the introduction section of the manuscript has been revised. The third paragraph has been broken into several parts according to the research categories in the revised manuscript as recommended by the respected reviewer. Thank you very much.

Comment-3: The major work and innovation need to be supplemented at the end of the introduction.

Response: The major work and innovations have been supplemented at the end of the introduction section in a point wise form. Please see the last paragraph in the Introduction Section. Thank you very much.

Comment-4: The symbols in text and formulas need to be unified. The font and format of the formulas need to be checked and modified. Authors should check the whole paper more carefully.

Response: The Authors apologies for such mistake and thank the respected reviewer for his remark. Kindly note that the authors have carefully checked the font and format of the symbols in the whole manuscript. Accordingly, the symbols used in the manuscript have been unified and modified in the revised manuscript. Thank you very much.

Comment-5: In the results part, the authors just showed the phenomenon of the experiments. The reason needs to be analyzed either. For example, in Boiling characteristics curves, there is no analysis of reason. The authors should explain the reason why the boiling curves of 0.01 and 0.1 turn towards the right.

Response: The authors have elaborated the results and discussion portion of the manuscript as per the recommendation of the respected reviewer. Thank you very much.

Comment-6: Figure 13 shows the contact angles at different concentrations. The contact angles seem to be measured before boiling. The contact angles measurement after the boiling should be supplemented. Otherwise, the effect of wettability cannot be confirmed. Generally, the hydrophilic surface shows better boiling performance.

Response: The authors cordially acknowledge the comment on contact angle. The authors like to state that the contact angle is measured before the testing of boiling phenomena in order to check its effect on boiling. However, the contact angle after boiling has not been measured since it was found that the silica nanofluids showed hydrophilicity and hence, it has been expected that contact angles after boiling will become more hydrophilic due to nanoparticle disposition. Furthermore, extensive study on the effect of contact angle and wettability on the pool boiling is kept as part of the future extension of the present work. Please see the last paragraph in the Conclusions Section. Thank you very much.

Comment-7: There is an error in figure 14. The value of the x-axis is wrong. Authors should check the whole paper carefully.

Response: The authors cordially acknowledge the comment on Figure 14. The x-axis of the graph was wrongly presented. The figure has been replaced with the correct one in the revised manuscript. Thank you very much.

Comment-8: The font and format errors are found on page 21.

Response: The authors apologies for such mistake to the respected reviewer. Kindly note that the font and format error in page 21 has been corrected in the revised manuscript. Thank you very much

Comment-9: The pictures of the bubbles need to be supplemented. The difference in bubble departure diameter needs to be explained.

Response: The authors have tried their best to improve the quality of the pictures of the bubbles in the revised version. The reason behind the difference in bubble diameter has been incorporated in the revision as recommended.

Finally, the authors would like to thank the respected reviewer for his time and very useful comments and remarks. Thank you very much.

Round 2

Reviewer 1 Report

Accept

Author Response

We thank the respected reviewer for sharing his time and knowledge with us.

Thank you very much

The Authors